# Region-Specific and Weather-Dependent Characteristics of the Relation between GNSS-Weighted Mean Temperature and Surface Temperature over China

**Minghua Wang** [1,2,3], **Junping Chen** [2,3,4,*] , **Jie Han** [1], **Yize Zhang** [2,3,4], **Mengtian Fan** [5] , **Miao Yu** [6] , **Chengzhi Sun** [1,7] **and Tao Xie** [1,7]

1   School of Remote Sensing and Geomatics Engineering, Nanjing University of Information Science and Technology, Nanjing 210044, China; mhwang@nuist.edu.cn (M.W.)

2   Shanghai Key Laboratory of Space Navigation and Positioning Techniques, Shanghai Astronomical Observatory, Chinese Academy of Sciences, Shanghai 200030, China

3   Shanghai Astronomical Observatory, Chinese Academy of Sciences, Shanghai 200030, China

4   University of Chinese Academy of Sciences, Beijing 100049, China

5   School of Geographical Sciences, Nanjing University of Information Science and Technology, Nanjing 210044, China

6   School of Atmospheric Sciences, Nanjing University of Information Science and Technology, Nanjing 210044, China

7   Technology Innovation Center for Integration Applications in Remote Sensing and Navigation, Ministry of Natural Resources, Nanjing 210044, China

\*   Correspondence: junping@shao.ac.cn

**Abstract:** Weighted mean temperature of the atmosphere, $T_m$, is a key parameter for retrieving the precipitable water vapor from Global Navigation Satellite System observations. It is commonly estimated by a linear model that relates to surface temperature $T_s$. However, the linear relationship between $T_m$ and $T_s$ is associated with geographic regions and affected by the weather. To better estimate the $T_m$ over China, we analyzed the region-specific and weather-dependent characteristics of this linear relationship using 860,054 radiosonde profiles from 88 Chinese stations between 2005 and 2018. The slope coefficients of site-specific linear models are 0.35~0.95, which generally reduce from northeast to southwest. Over southwest China, the slope coefficient changes drastically, while over the northwest, it shows little variation. We developed a $T_s \sim T_m$ linear model using the data from rainless days as well as a model using the data from rainy days for each station. At half the stations, mostly located in west and north China, the differences between the rainy-day and rainless-day $T_m$ models are significant and larger than 0.5% (1%) in mean (maximal) relative bias. The regression precisions of the rainy-day models are higher than that of the rainless-day models averagely by 28% for the stations. Radiosonde data satisfying $T_m - T_s > 10$ K and $T_s - T_m > 30$ K most deviate from linear regression models. Results suggest that the former situation is related to low surface temperature (<270 K), as well as striking temperature and humidity inversions below 800 hPa, while the latter situation is related to high surface temperature (>280 K) and a distinct humidity inversion above 600 hPa.

**Keywords:** GNSS meteorology; weighted mean temperature of atmosphere; linear relation; weather dependence; site-specific model; radiosonde data

## 1. Introduction

Retrieval of precipitable water vapor (PWV) from Global Navigation Satellite System (GNSS) observations is a major work of ground-based GNSS meteorology [1–4]. This technique has all-weather and all-time capabilities, and can provide water vapor products with high spatiotemporal resolution as the continuous GNSS observing network grows and the data processing strategy improves [5]. The GNSS-derived PWV are consistent

with the radiosonde and ERA5 data [6,7]. Nowadays, the use of ground-based GNSS, effectively complementing the conventional water vapor observations, plays an important role in weather and climate studies, such as water vapor variation [8,9], satellite product validation [10,11], deep convection and rainstorm observation [12–14], and monsoon and atmospheric river monitoring [15,16]. In GNSS water vapor retrieving, a crucial step is to convert zenith wet delay (ZWD) into the PWV [17]. The conversion factor between ZWD and PWV is a function of the weighted mean temperature of the atmosphere (hereinafter referred to as $T_m$), which is a quantity defined by Davis et al. [18].

It is worthwhile to obtain the $T_m$ as accurate as possible since the uncertainty of the $T_m$ is the dominant error source affecting the conversion between ZWD and PWV [19]. Since the vertical profiles of temperature and humidity usually cannot be accessed at the GNSS stations, the calculation of the $T_m$ from the integral equation by Davis et al. [18] is not always feasible in practical applications. To overcome this, Bevis et al. [1] took advantage of the significant linear correlation between $T_m$ and surface temperature (referred to as $T_s$) to develop a linear model ($T_m = 0.72T_s + 70.2$) from the radiosonde profiles observed over U.S. continent. This linear model has been used widely due to its easy implementation. Although the Bevis model is in essence a regional model, it was used as a global model [20–22] or as the reference model to validate other global or regional models [23–25]. Nevertheless, Ross and Rosenfeld [26] showed that the linear relationship between $T_m$ and $T_s$ varied with geographic location, indicating that a single linear model cannot assure high accuracy of $T_m$ calculation for the whole globe. Wang et al. [27] evaluated the Bevis model using the global ERA-40 data and found that the mean bias of $T_m$ generated from this model has a range of $\pm 10$ K (relative bias of $\pm 3.5\%$). To obtain accurate $T_m$, geodesists and meteorologists developed regional models (with form similar to the Bevis model or a little more complicated) for their concerned regions, e.g., Netherlands [28], Indian [24,29], Brazil [30], Australia [31], European region [8,32], West Africa [33], Greenland [34], Taiwan [35], and Hong Kong [36,37]. Additionally, there are some empirical models that only require the inputs of location and time to estimate the $T_m$ values. The representatives of these models are the GPT model series, GPT2w [38] and GPT3 [39], and the GTm model series, GTm-I [40], GTm-II [21], and GTm-III [41]. These $T_s$-independent empirical models are very useful especially for those GNSS stations without surface temperature observations. However, when the surface temperatures are available, $T_s$-dependent models are better options because that they generally yield more accurate $T_m$ values than the $T_s$-independent models [23,32,42,43].

During the past two decades, the number of GNSS stations grew rapidly in China. The Meteorological Observation Center, China Meteorological Administration collects the data of more than 1000 GNSS stations covering the whole mainland China, and calculates the PWV values on a daily basis serving for weather analysis, forecasting, and scientific studies [44]. These stations are mostly from China Meteorological Administration GNSS Network (CMAGN) and Crustal Movement Observation Network of China (CMONOC), and equipped with meteorological sensors that record the surface pressure and temperature for GNSS PWV retrieving. Since the surface temperature is available, the simple linear $T_s \sim T_m$ models can be used to calculate $T_m$. However, China is a geographically large country with a variety of climate regions. Previous studies [45–48] showed the coefficients of regional $T_s \sim T_m$ linear models for different areas in China are more or less different. This motivates us to comprehensively study and obtain the accurate linear relationships between $T_m$ and $T_s$ over China with the purpose of providing the basis for high accuracy $T_m$ estimation in the region.

In this study, using 14-year data from 88 Chinese radiosonde stations, we developed a unified $T_s \sim T_m$ linear model for the whole China as well as site-specific linear models for individual stations, and calculated the representativeness errors of the unified model relative to the site-specific models. Then, we analyzed the variation of the linear relation between $T_m$ and $T_s$ with geographic locations and different weather occurrences. We also

investigated the regression precision of the generated $T_s \sim T_m$ linear models and the weather conditions related to the data that most deviate from the regressions.

## 2. Methods and Data

### 2.1. Role of $T_m$ in GNSS Water Vapor Retrieving

Before the determination of $T_s \sim T_m$ linear relations with radiosonde data, we briefly review the role of $T_m$ in the retrieval of water vapor from GNSS observations. In GNSS high-precision positioning, especially when the precise point positioning (PPP) is applied, the zenith tropospheric delay (ZTD) of L-band signals over a station is estimated simultaneously with the coordinate components. The ZTD includes two parts: zenith hydrostatic delay (ZHD) and ZWD. With a known surface pressure, the ZHD can be estimated with an accuracy of millimeter level. Subtracting the ZHD from the ZTD remains the ZWD. The PWV is obtained from ZWD as [19]

$$\text{PWV} = \Pi \cdot (\text{ZTD} - \text{ZHD}) = \Pi \cdot \text{ZWD} \tag{1}$$

where $\Pi$ is a dimensionless mapping factor. Its expression is

$$\Pi = \frac{10^6}{\rho R_v \left( \frac{k_3}{T_m} + k_2' \right)} \tag{2}$$

where $\rho$ is the density of liquid water, $R_v$ is the specific gas constant for water vapor, and $k_2'$ and $k_3$ are constants that have been evaluated by the actual measurement of refractivity index of the atmosphere [19,49]. Since $\rho$, $R_v$, $k_2'$, and $k_3$ are all known constants, $T_m$ is the only quantity to be determined for the conversion of ZWD into PWV.

### 2.2. Determination of $T_s \sim T_m$ Linear Models

The $T_s \sim T_m$ linear model is expressed as

$$T_m = aT_s + b \tag{3}$$

where all temperatures are in kelvins. $a$ (slope) and $b$ (intercept) are regression coefficients. To determine $a$ and $b$ in regression analysis, a number of $T_s \sim T_m$ data pairs are required to be known. Due to global distribution of sites and long-term data accumulation, radiosonde is a good data source to derive $T_s$ and $T_m$ for establishing $T_s \sim T_m$ linear models over land.

Surface temperatures are obtained from the first level of radiosonde profiles, and weighted mean temperatures are calculated from water vapor pressures and temperatures at all levels. The definition of $T_m$ is [18]

$$T_m = \frac{\int_{H_S}^{H_T} \frac{P_w}{T} dH}{\int_{H_S}^{H_T} \frac{P_w}{T^2} dH} \tag{4}$$

where $P_w$ is the partial pressure of water vapor, $T$ is the temperature of the atmosphere, $H_T$ is the height of the top of the atmosphere, and $H_S$ is the starting altitude. For GNSS meteorology, $H_S$ is the height of GNSS antenna. When calculating $T_m$ from radiosonde profiles, the discrete form of Equation (4) is used as

$$T_m = \frac{\sum\limits_{i=1}^{N} \frac{P_{wi}}{T_i} \Delta H_i}{\sum\limits_{i=1}^{N} \frac{P_{wi}}{T_i^2} \Delta H_i} \tag{5}$$

where $N$ is the total number of layers of the atmosphere with one layer defined as the atmosphere between two consecutive levels of a radiosonde profile, $T_i$ is the average temperature in the $i$ layer, $\Delta H_i$ is the thickness of the $i$ layer, and $P_{wi}$ is the average water

vapor pressure in the *i* layer. Generally, the radiosonde profiles do not directly provide partial pressure of water vapor $P_w$, and instead provide total atmospheric pressure $P$ and mixing ratio of water vapor $m_x$, from which the $P_w$ can be derived by [50].

$$P_w = \frac{m_x}{m_x + 622} P \tag{6}$$

where $m_x$ is in g/kg.

### 2.3. Radiosonde Data

The radiosonde data of 88 stations selected from 2005 to 2018 (14 years) are used to analyze the $T_s \sim T_m$ relations over China. Figure 1 shows the geographic distribution of the radiosonde stations and the topography. The station information is shown in Table A1 of Appendix A. For these stations, most radiosonde observations are taken at 00:00 UTC and 12:00 UTC daily. All radiosonde data were downloaded from the upper air sounding archive of University of Wyoming (http://weather.uwyo.edu/upperair/sounding.html, accessed on 1 May 2022 ). At most stations, we used the data from 2005 to 2018 for the analyses, while at station ZHANGQIU (ID: 54727), WENJIANG (ID: 56187), and JINGHE (ID: 57131), we only used the data from 2014 to 2018 because the site information is incomplete in data files from 2005 to 2013 for the three stations.

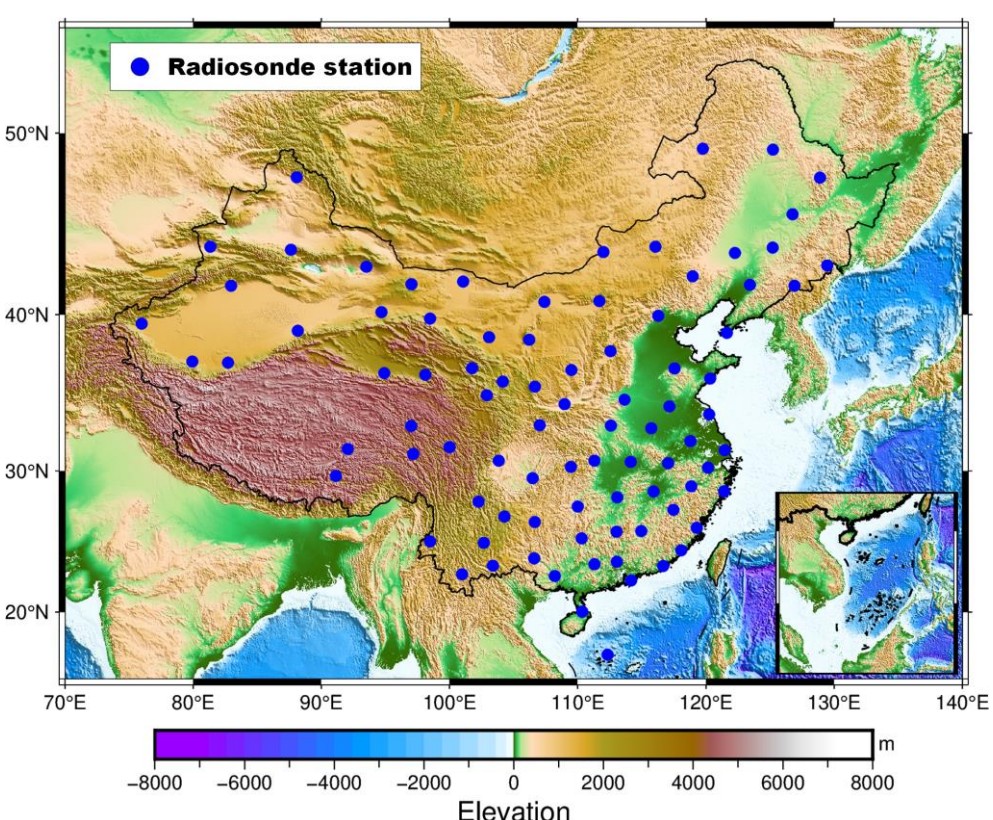

**Figure 1.** The geographic distribution of 88 selected radiosonde stations over China. The blue dots denote the locations of the radiosonde stations and the color on the map represents the elevation. The map is generated by Generic Mapping Tools [51], and the 2-min Gridded Global Relief Data (ETOPO2) v2 are used.

Before data processing, we carried out the quality checking for each radiosonde profile. An accepted radiosonde profile is required to contain the data of the first level, which is the level at the altitude of that station. The pressure of the top level is required to be no more than 300 hPa. In addition, the profile must contain the standard pressure levels and the total number of levels should be no less than 5.

For most radiosonde data, $T_m$ is less than $T_s$, but the difference $(T_s - T_m)$ is normally no more than 30 K. In order to detect the profiles with gross error, we manually checked both the profiles with $T_s - T_m > 30$ K and $T_m - T_s > 10$ K. Among 724 checked radiosonde profiles, 39 of them were found to contain obvious record errors or unreasonable temperature gradients, and they were then removed from the later data processing.

## 3. Unified Model

The Bevis model $T_m = 0.72T_s + 70.2$ was developed using 8712 radiosonde profiles at 13 U.S. stations between 1989 and 1991, and the root mean square (RMS) deviation from the regression is 4.74 K [1]. In this study, we used much more data, 860,054 profiles at 88 stations from 2005 to 2018, to establish a $T_s \sim T_m$ linear model over China (Figure 2), hereinafter referred to as the unified model. The unified model is $T_m = 0.79T_s + 50.76$, with an RMS about the regression of 4.14 K (14% smaller than the RMS for the Bevis model).

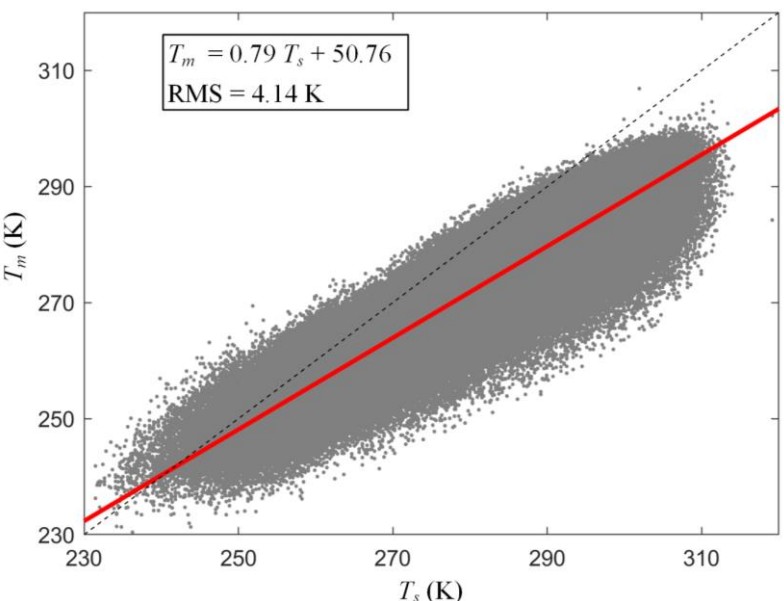

**Figure 2.** Relationship between $T_m$ and $T_s$ determined from 860,054 radiosonde profiles over China. The radiosonde profiles are acquired from the 88 stations (Figure 1) between 2005 and 2018. The red solid line is the linear regression result of the data (the unified model), and the black dotted line shows the positions where $T_m$ and $T_s$ are equal.

Though using a single $T_s \sim T_m$ linear model to estimate $T_m$ for a large region is convenient in practice, it may introduce large representativeness errors. Ross and Rosenfeld [26] pointed out that, over the U.S., the slope of the Bevis model (0.72) is not that representative of the slopes of site-specific models. Like the U.S., China also has a vast territory, and thus the simply unified model for the whole China may cause significant $T_m$ estimation errors as well. We developed a $T_s \sim T_m$ linear model for each single station and compared the unified model with the 88 site-specific models. The slope $a$ and intercept $b$ of each site-specific model are shown in columns 5 and 6 of Table A1 (Appendix A), respectively. Figure 3 illustrates that some site-specific models show good consistency with the unified model, while others show clear deviation from the unified model. In some situations, the difference between $T_m$ from the unified model and that from some site-specific model reaches >10 K, which is equivalent to 3% $T_m$ relative errors.

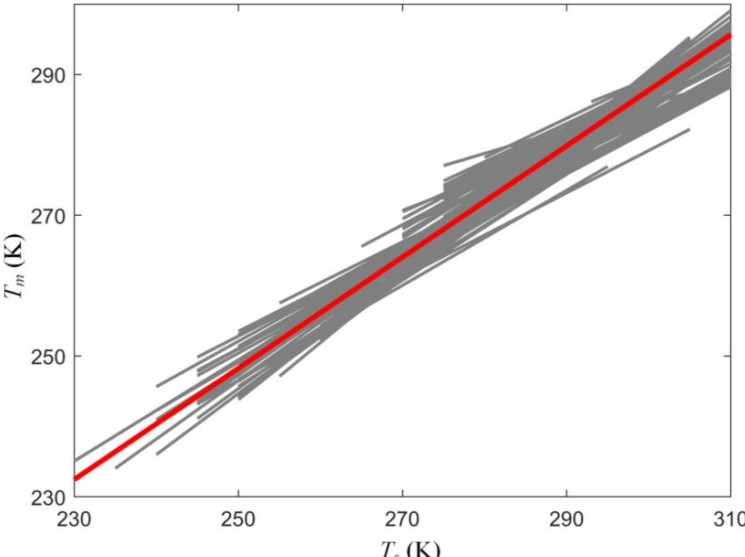

**Figure 3.** The unified $T_s \sim T_m$ linear model (red line) and site-specific models (gray lines). The range of $T_s$ for a site-specific model describes the surface temperatures that the station real observed.

To quantify the differences between the unified model and each site-specific model, we calculated the $T_m$ biases ($bias = |T_{m\_}Unified - T_{m\_}Site|$) between them. Figure 4 shows the $T_m$ mean bias, maximal bias, mean relative bias, and maximal relative bias of the unified model relative to each site-specific model. The mean bias is larger than 2 K at 45 stations (51% of all the stations) (Figure 4a), and the maximal bias is larger than 4 K at 46 stations (52%) (Figure 4b). In terms of relative bias, at some stations, the mean relative bias is close to or above 1.5% (Figure 4c) and the maximal relative bias is over 3% (Figure 4d). At more than half the stations, the mean relative bias is larger than 0.5% or the maximal relative bias is larger than 1%. All these results suggest that the unified model is not a good proxy for more than half the 88 site-specific models if the 2 K mean bias, 4 K maximal bias, 0.5% mean relative bias, or 1% maximal relative bias is chosen as the threshold.

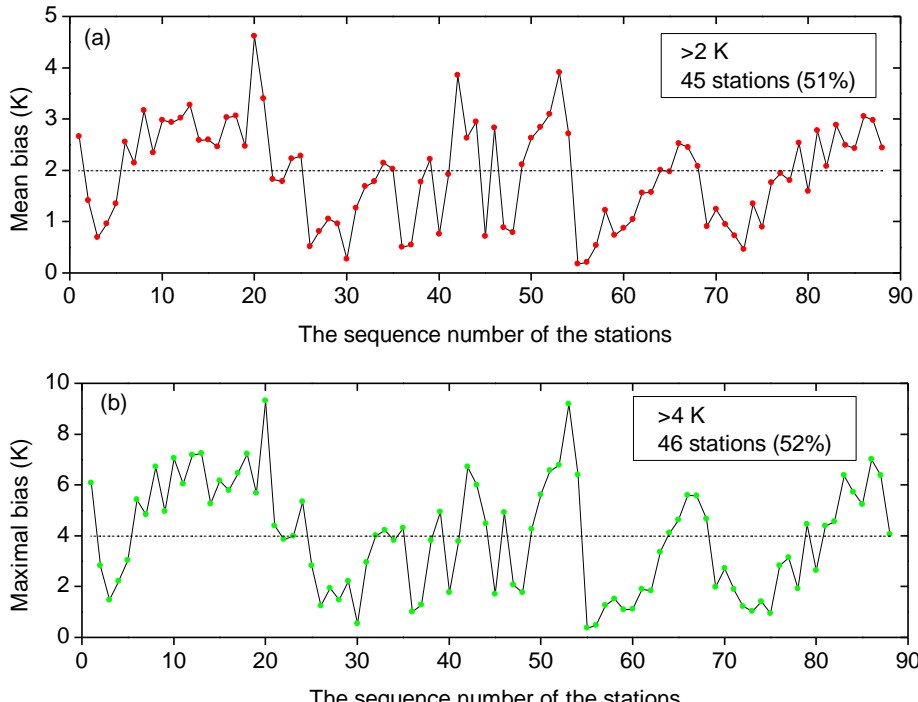

**Figure 4.** *Cont.*

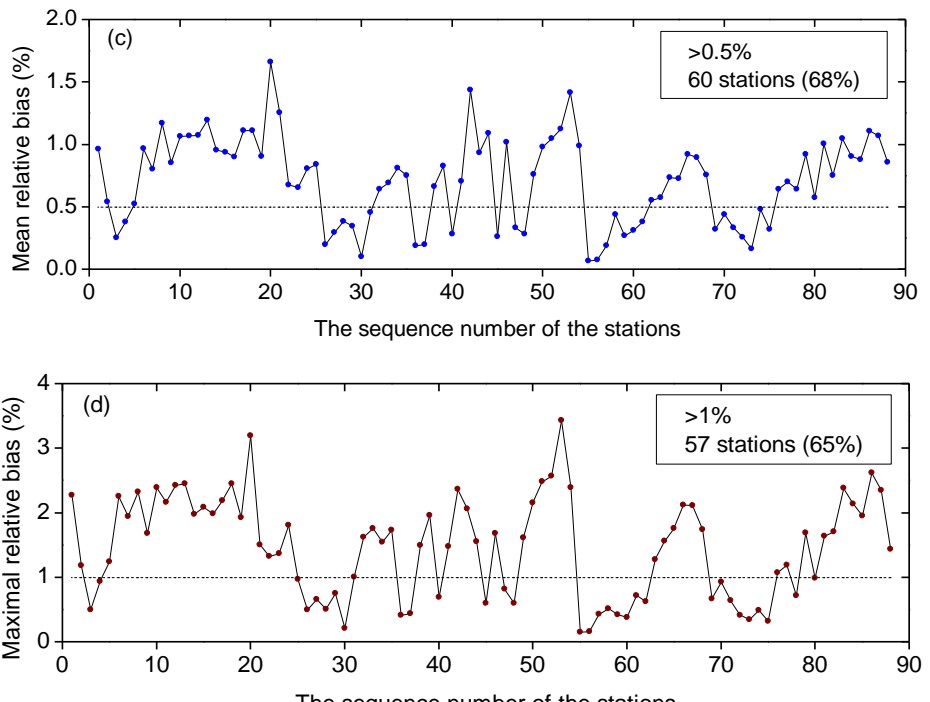

**Figure 4.** Differences between the unified model and each site-specific model. The bias is the absolute value of the difference between unified-model $T_m$ and site-specific-model $T_m$. The relative bias is defined as the bias divided by the site-specific-model $T_m$ value. For each station, the mean (**a**) and maximum (**b**) of the biases, as well as the mean (**c**) and maximum (**d**) of the relative biases, are shown. The horizontal axis of each plot is the sequence number of the radiosonde stations. The correspondences between the sequence numbers and station names are shown in Table A1 of Appendix A (columns 1 and 2). NO. 1 represents the station Kings Park at Hong Kong. NO. 2–88 represent the stations located at Inland China, and these stations are arranged from high latitude to low latitude.

## 4. Region-Specific Characteristics of $T_s \sim T_m$ Linear Relations

The distinct deviation of the unified model from part of the site-specific models lies in the diversity of the $T_s \sim T_m$ linear relations at different areas. We plotted the contours of the slopes (coefficient *a*) of site-specific $T_s \sim T_m$ linear models (Figure 5a), which shows the slope generally increasing from low to middle latitudes. Figure 5a shows that the range of slopes is from 0.35 to 0.96 over China, with large slopes (>0.9) occurring at Bohai bay and small slopes (<0.5) at the south of Yunnan (YN) province (refer to Figure 5b for the area names and their locations). In general, the slope decreases from the northeast to the southwest. In the area of Xinjiang (XJ) and the west of Neimenggu (NM), Gansu (GS), Qinghai (QH), and Xizang (XZ), the slope changes gently. Thus, applying one linear model to the whole area does not give rise to significant representativeness errors. Another similar area where the slope also shows little variation includes Shaanxi (SA), the west of Shanxi (SX), and the north of Sichuan (SC). In contrast, the slope changes drastically over the area of Yunnan (YN) and the border between Sichuan (SC) and Guizhou (GZ), which indicates that using a single $T_s \sim T_m$ linear model for this area will inevitably result in large representativeness errors.

The regression precision of site-specific $T_s \sim T_m$ linear model is related to site location. To investigate the relation between regression precisions of the models and the geographic location, we calculated the RMS deviation from the regression of site-specific model for each station. Figure 6 shows that the stations (TENGCHONG and KINGS PARK) with small RMS (<2 K) are located at low latitudes, while those (DUNHUANG and EJIN QI)

with large RMS (>5 K) are located at relative high latitudes. Overall, the RMS tends to increase with latitude.

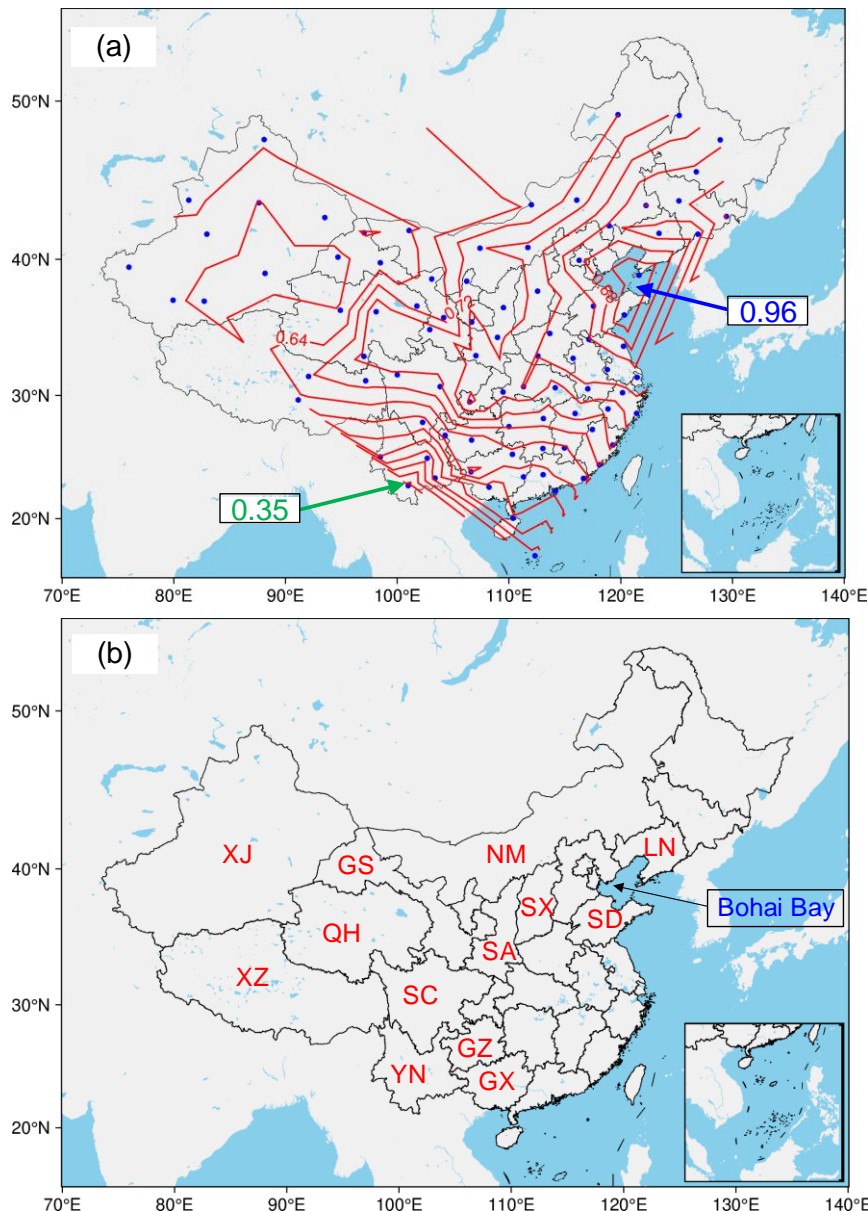

**Figure 5.** Slopes of $T_s \sim T_m$ linear regression models. (**a**) the contour lines of the slopes of the linear models. The contour interval is 0.04. The plot (**b**) provides an area name and the names of provinces (acronym) that are referred to in the text.

To better understand the variation of the regression precision with latitude, the distributions of radiosonde $T_s \sim T_m$ data points from two stations with low latitude (TENGCHONG and KINGS PARK) and two stations with relative high latitude (DUNHUANG and EJIN QI) are compared in Figure 7 (refer to Figure 6 for the locations of the four stations). In both the $T_s$-axis and $T_m$-axis dimensions, the data points from high-latitude stations, EJIN QI (Figure 7c) and DUNHUANG (Figure 7d), are much more dispersedly distributed than those from low-latitude stations, KINGS PARK (Figure 7a) and TENGCHONG (Figure 7b). The ranges of $T_s$ at station KINGS PARK and TENGCHONG are much smaller than those at EJIN QI and DUNHUANG. This is because the surface temperatures are relative high at low latitudes all year round, and thus they concentrate on a smaller range. For a certain $T_s$, the variations of $T_m$ at stations with low latitudes are also smaller. A closer inspection of

the radiosonde profiles suggests that, in general, the vertical distributions of water vapor and temperature are more uniform for low-latitude stations than for high-latitude stations (not shown), which accounts for the smaller $T_m$ variation at low latitudes.

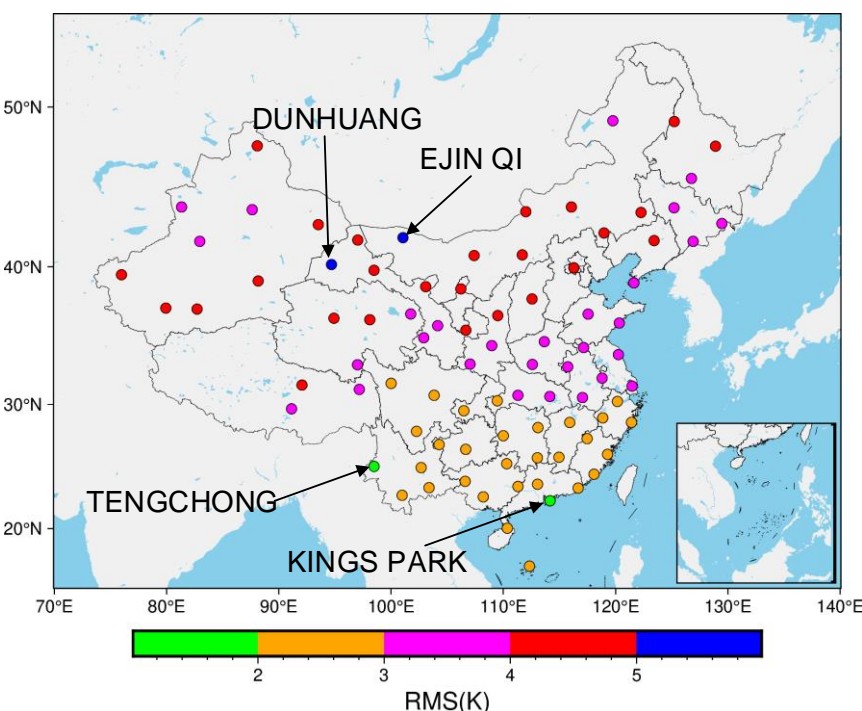

**Figure 6.** Root mean squares of the 88 site-specific $T_s \sim T_m$ linear regression models.

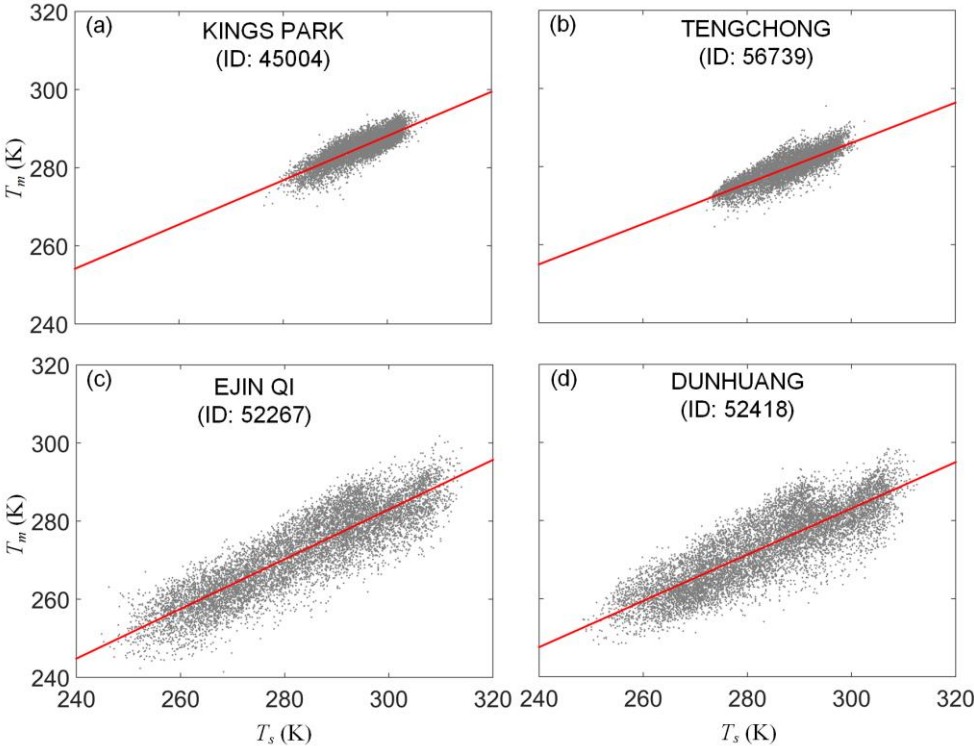

**Figure 7.** Radiosonde data points (gray dots) and $T_s \sim T_m$ linear regression models (red lines) at the station (**a**) Kings Park, (**b**) Tengchong, (**c**) Ejin Qi, and (**d**) Dunhuang. There are 10,054 profiles used at Kings Park, 9933 profiles at Tengchong, 9568 profiles at Ejin Qi, and 9907 profiles at Dunhuang.

## 5. Weather-Dependent Characteristics of $T_s \sim T_m$ Linear Relations

Under different kinds of weather, the vertical distributions of temperature and water vapor can vary considerably. As a result, the $T_s \sim T_m$ relation is likely to change accordingly. For each station, we used the radiosonde data from different weather separately to generate weather-dependent $T_s \sim T_m$ linear models. We then compared the weather-dependent $T_s \sim T_m$ linear models as well as their regression precision. In some $T_s \sim T_m$ plots, such as Figures 2 and 7, some data points are far away from the regression line. We investigated two sets of these points and analyzed their related weather conditions.

### 5.1. Weather-Dependent $T_s \sim T_m$ Linear Models

For simplification, we only consider two kinds of weather: rain and no rain. All the days involved in the experiment are classified as either a rainy day or a rainless day. If there is no rain for a whole day, that day is defined as a rainless day, and if not, the day is defined as a rainy day. We used the daily precipitation data acquired from the data center of China Meteorology Administration (http://data.cma.cn/, accessed on 16 May 2022) to classify the days. For each station, we generated a $T_s \sim T_m$ linear model using the radiosonde data from rainless days and a model using the data from rainy days. The former model is hereinafter referred to as "rainless-day model" and the latter is referred to as "rainy-day model". Table A1 of Appendix A shows the slope $a$ and intercept $b$ of the rainless-day models (columns 7 and 8) and rainy-day models (columns 9 and 10) for all the 88 stations.

At some stations, such as station NAGQU, XICHANG, KUNMING, and SIMAO shown in Figure 8, the rainy-day model clearly deviates from the rainless-day model. The difference between the $T_m$ from rainy-day model and that from rainless-day model can be larger than 5 K. Thus, in order to get higher accuracy $T_m$ values for these stations, it is better to use the rainy-day model for rainy days and the rainless-day model for rainless days rather than using a weather-independent model for all days. Meanwhile, at some other stations, such as station HARBIN, BEIJING, WUHAN, and SHANGHAI shown in Figure 9, the rainy-day model shows good consistency with the rainless-day model. Hence, for these stations, it is not necessary to use weather-dependent models for rainy days and rainless days separately.

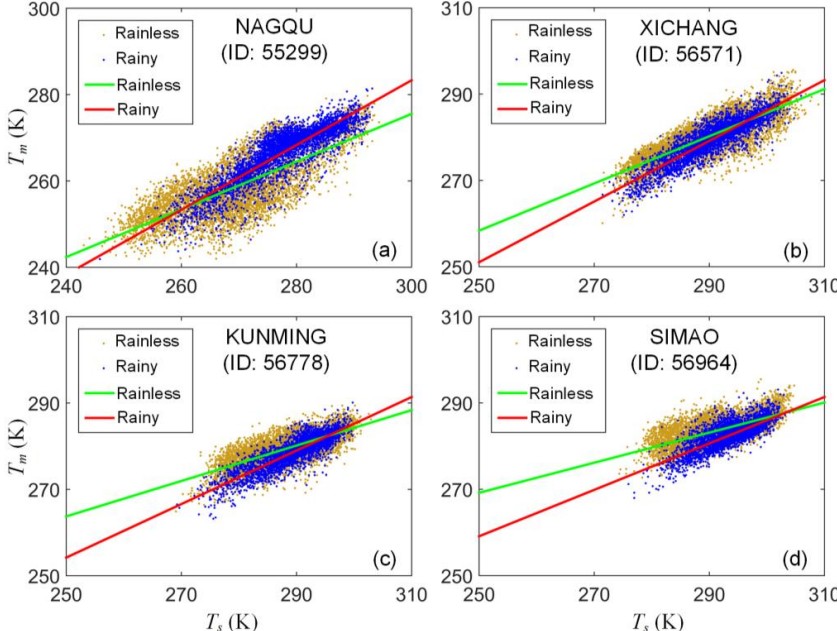

**Figure 8.** Rainy-day model and rainless-day model for station (**a**) NAGQU, (**b**) XICHANG, (**c**) KUNMING, and (**d**) SIMAO. The green lines are the rainless-day models and the red lines are the rainy-day models. The golden dots are the radiosonde data from rainless days and the blue dots are the data from rainy days.

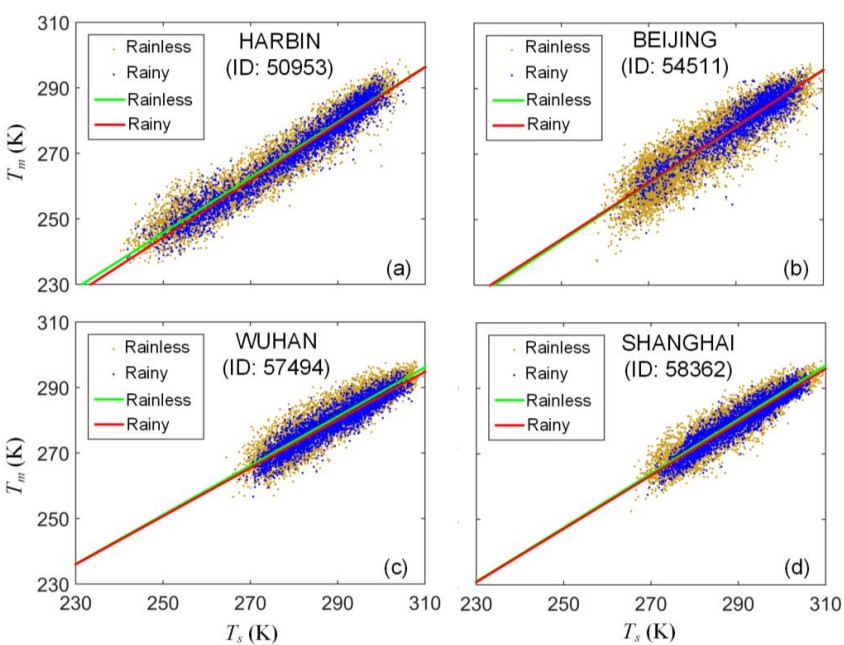

**Figure 9.** Rainy-day model and rainless-day model for station (**a**) HARBIN, (**b**) BEIJING, (**c**) WUHAN, and (**d**) SHANGHAI. The meanings of the lines and dots are the same as those in Figure 8.

We compared the rainy-day model with the rainless-day model for each station. At half the stations, the difference between the two models is significant: the mean relative bias of $T_m$ between the two models is larger than 0.5%, or the maximal relative bias is larger than 1%. Figure 10 shows that the stations with significant difference between the rainy-day model and rainless-day model (red dots in Figure 10) distribute over a larger area than the stations with insignificant model difference (blue dots in Figure 10). In general, the stations with significant model difference are at higher altitudes (refer to Figure 1 for the topography), and they are mostly located in the west and north of China. This suggests that, in general, using weather-dependent models for rainy and rainless days separately will effectively improve the $T_m$ accuracy in the west and north of China, but not in the eastern part.

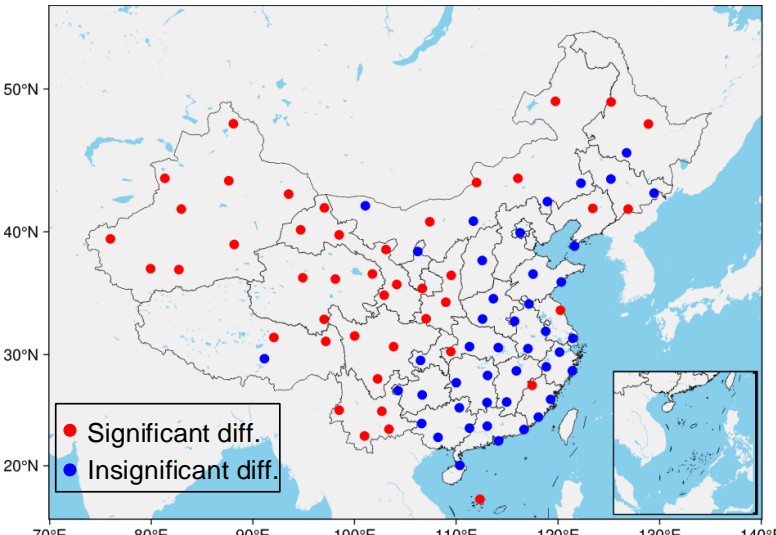

**Figure 10.** Stations with significant (red) and insignificant (blue) difference between the rainy-day model and the rainless-day model. If the mean relative bias of $T_m$ between the rainy-day model and the rainless-day model is larger than 0.5% or the maximal relative bias of $T_m$ between them is larger than 1%, it is classified as significant difference, and if not, it is insignificant difference.

### 5.2. Comparison of Regression Precision of Weather-Dependent Models

Both Figures 8 and 9 show that the distributions of data points from rainy days (blue dots) are more concentrated than those from rainless days (golden dots), indicating that the temperature and water vapor vertical profiles are more uniform on rainy days than on rainless days. We calculated the root mean squares deviation from the regressions of the weather-dependent models. Figure 11 shows that the RMS for rainy-day model is less than that for rainless-day model at all the stations, and the reduction rate of the rainy-day model RMS relative to the rainless-day model RMS is from 4% to 49% (average: 28%). This result suggests that the rainy-day model yielded $T_m$ values for rainy days are expected to have better accuracy than the rainless-day model yielded ones for rainless days.

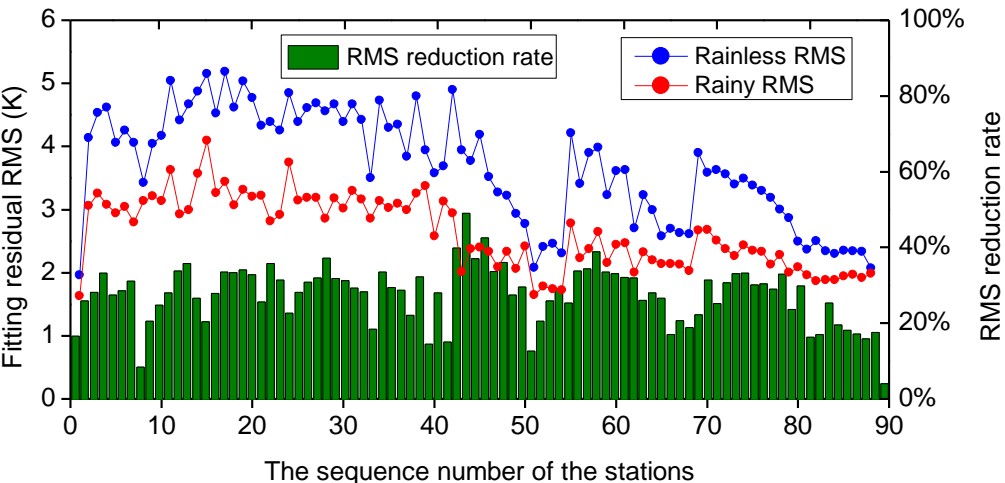

**Figure 11.** Regression precision of the rainy-day model and the rainless-day model for each station. The correspondence between the sequence numbers (horizontal axis) and station names is shown in Table A1 of Appendix A (columns 1 and 2). The red dot (blue dot) represents the RMS about the regression of rainy-day model (rainless-day model). The green bar denotes the reduction rate of rainy-day model RMS relative to the rainless-day model RMS.

While using the weather-dependent models for rainy days and rainless days separately can improve the accuracy of $T_m$ estimates, the benefits for rainy days and rainless days are unequal. Since the number of radiosonde profiles from rainless days is more than that from rainy days for the stations, the weather-independent model of a station is closer to the rainless-day model than to the rainy-day model. If the weather-independent model is used, its representativeness error for the rainy-day model is larger than that for the rainless-day model. Hence, replacing the weather-independent model with the weather-dependent models results in more improvement in the accuracy of $T_m$ estimates for rainy days than for rainless days.

### 5.3. Weather Conditions Related to Some Specific Data Points

In most cases, as shown in Figure 2, $T_m$ is less than $T_s$, and $T_s$ minus $T_m$ is generally no more than 30 K. For the cases of $T_m$ larger than $T_s$, $T_m$ minus $T_s$ is normally no more than 10 K. However, in some situations, extremes of $T_m - T_s > 10$ K and $T_s - T_m > 30$ K happen. Figure 12 shows the data points from the radiosonde profiles corresponding to the extremes. These data points are the ones that deviate most from the unified $T_s \sim T_m$ linear model, and they are responsible for the large RMS of the linear regression. To better understand the cause of the extremes, the weather conditions related to these specific data points are investigated.

Figure 12 shows that, in the dataset of $T_m - T_s > 10$ K and $T_s - T_m > 30$ K, the data from rainy days are much less than that from rainless days: the former only accounts for 4%. Such a small proportion of data points from rainy days suggests that the situations

of $T_m - T_s > 10$ K and $T_s - T_m > 30$ K, which reduce the precision of the $T_s \sim T_m$ linear regression, seldom occur on rainy days or mostly happen on rainless days. This result partly explains why the rainy-day model has better regression precision than the rainless-day model.

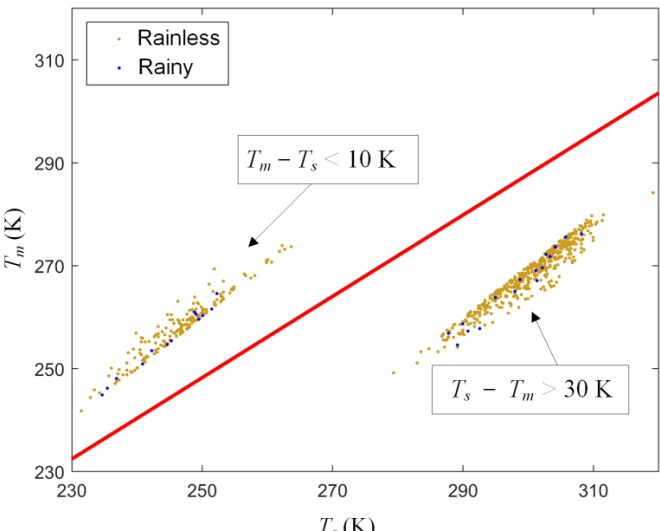

**Figure 12.** Data points from the radiosonde profiles that satisfy $T_m - T_s > 10$ K and $T_s - T_m > 30$ K. The red line is the unified $T_s \sim T_m$ linear model, the same as that shown in Figure 2. The golden dots are the data from rainless days and the blue dots are those from rainy days.

The statistics suggests that 98% of the radiosonde observations with $T_m - T_s > 10$ K were in winter, and 92% of them were observed at 00:00 UTC, while for the situation of $T_s - T_m > 30$ K, 97% of the observations were in summer and spring, and 98% of them were observed at 12:00 UTC. For stations in China, the surface temperature is normally low at 00:00 UTC (8:00 Beijing time) in winter, while at 12:00 UTC (20:00 Beijing time) in summer and spring, the surface temperature is relatively high. This indicates that the situation of $T_m - T_s > 10$ K occurs on the condition of low surface temperature, while the situation of $T_s - T_m > 30$ K happens under the circumstance of high surface temperature. This conclusion is confirmed by Figure 12, which shows the data points with $T_m - T_s > 10$ K are all distributed in $T_s < 270$ K, and those with $T_s - T_m > 30$ K are all in the range of $T_s > 280$ K.

To further investigate the weather backgrounds related to the extremes of $T_m - T_s > 10$ K and $T_s - T_m > 30$ K, we checked the temperature and water vapor mixing ratio profiles for the data points corresponding to these extremes. All data with $T_m - T_s > 10$ K have a similar vertical distribution of temperature and also a similar vertical distribution of the mixing ratio. Figure 13 shows the representative temperature and mixing ratio profiles for the situation of $T_m - T_s > 10$ K. This situation is related to a temperature inversion and a humidity inversion. The temperature (humidity) inversion is a weather phenomenon that the temperature (humidity) increases with the height [52]. In these profiles, the temperature inversion occurs at a similar height as the humidity inversion, and both of them are striking and are below 800 hPa.

While for the data with $T_s - T_m > 30$ K, all temperature (mixing ratio) profiles show a similar vertical distribution, but the vertical distribution is clearly different from that of the data with $T_m - T_s > 10$ K. Figure 14 shows the representative temperature and mixing ratio profiles for the situation of $T_s - T_m > 30$ K. The profiles show that there is a distinct humidity inversion in each mixing ratio profile, but no obvious temperature inversion occurs at any observation height. For the data with $T_s - T_m > 30$ K, the height of humidity inversion is generally above 600 hPa, which is much higher than the height of humidity inversion for the data with $T_m - T_s > 10$ K.

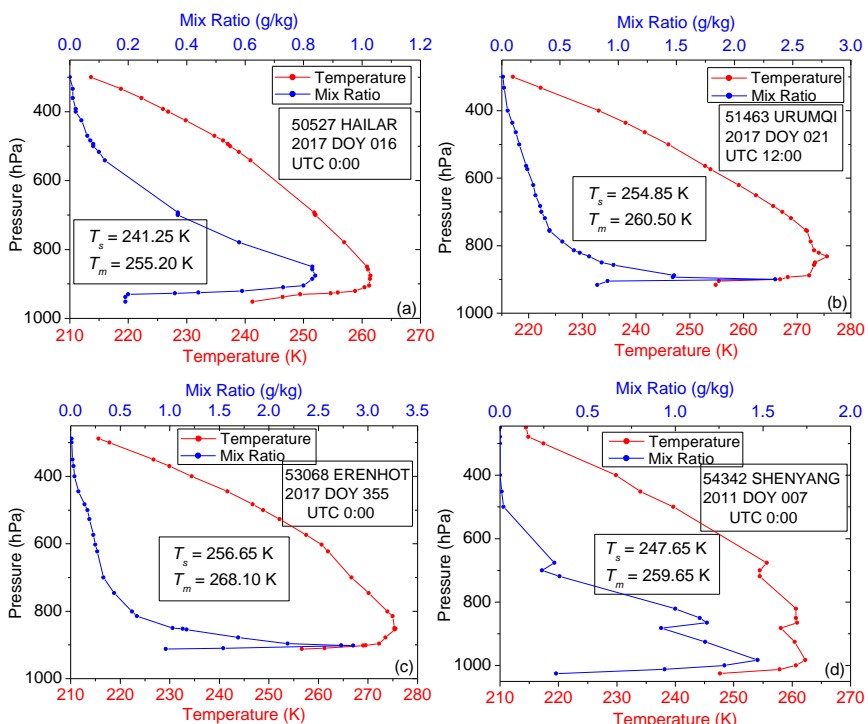

**Figure 13.** Representative vertical profiles of temperature (red) and water vapor mixing ratio (blue) from radiosonde observations that satisfy $T_m - T_s > 10$ K at station (**a**) HAILAR, (**b**) URUMQI, (**c**) ERENHOT, and (**d**) SHENYANG. The radiosonde observing time as well as $T_s$ and $T_m$ are shown in each panel.

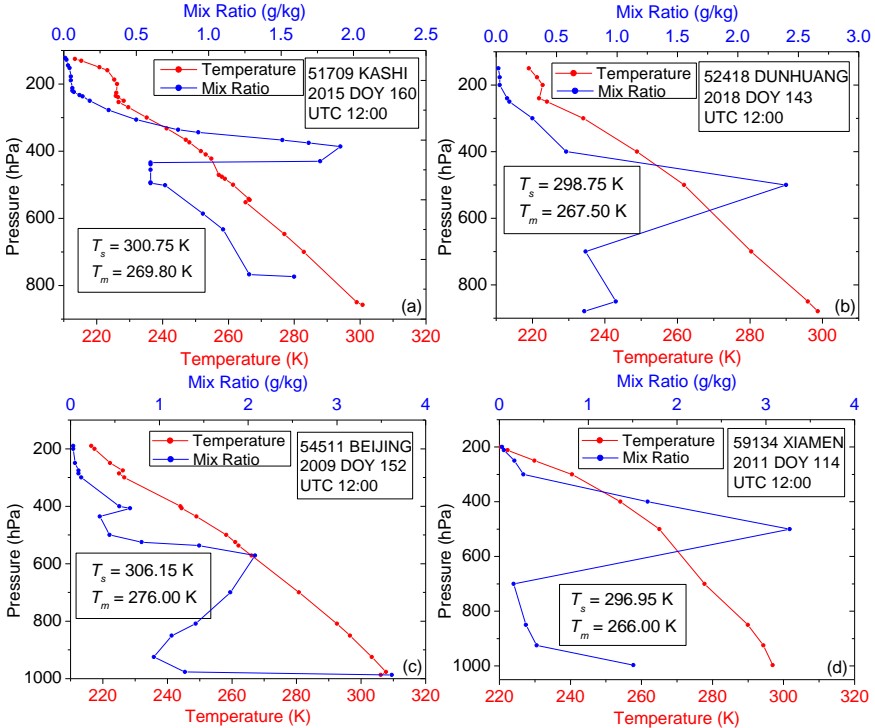

**Figure 14.** Representative vertical profiles of temperature (red) and water vapor mixing ratio (blue) from radiosonde observations that satisfy $T_s - T_m > 30$ K at station (**a**) KASHI, (**b**) DUNHUANG, (**c**) BEIJING, and (**d**) XIAMEN. The radiosonde observing time as well as $T_s$ and $T_m$ are shown in each panel.

## 6. Conclusions

Analysis of 14 years of radiosonde profiles at 88 stations demonstrates that the $T_s \sim T_m$ linear relation is region specific over China. Using a single $T_s \sim T_m$ linear model to represent all the 88 site-specific models for estimating $T_m$ over the area of the whole of China brings diverse errors of up to 10 K (about 3% relative error), indicating that the region-specific characteristics of the $T_s \sim T_m$ linear relations must be considered in order to accurately estimate $T_m$. The geographical distribution of the slopes (coefficient $a$ of $T_m = aT_s + b$) of the site-specific models reflects the variation of the $T_s \sim T_m$ linear relations over different areas. From Bohai bay to the south of Yunnan province, the slope reduces from 0.96 to 0.35. Over Yunnan and the border between Sichuan and Guizhou, the slope changes drastically, while over some other areas, such as Xinjiang and part of Qinghai, the slopes change slowly. This information provides the basis for determining which area can simply use a single $T_s \sim T_m$ linear model to estimate the $T_m$ without bringing in significant representativeness errors and which areas have to use multiple models for highly accurate $T_m$ estimation. The characteristic of slope increasing with latitude can also be found from the study of Ross and Rosenfeld [26] and Yao et al. [53]. However, their work was on a global scale and did not provide detailed analysis specifically for China as this study does. The regression precision of a site-specific $T_s \sim T_m$ linear model is also related to the geographical location. In general, the RMS about the regression of the site-specific model increases with station latitude, and this conclusion is similar to the result of Raju et al. [29] for the Indian subcontinent.

Investigation of the $T_s \sim T_m$ linear models generated from the radiosonde data observed on rainy days (rainy-day model) and on rainless days (rainless-day model) demonstrates that the $T_s \sim T_m$ linear relation is weather dependent. At half the stations, the difference between the $T_m$ estimates from the rainy-day model and those from the rainless-day model is significant (mean relative bias larger than 0.5% or maximal relative bias larger than 1%). These stations are mostly located in the west and north of China. Thus, over these regions, both the region-specific and weather-dependent characteristics of the $T_s \sim T_m$ linear relations should be taken into account for ensuring the accuracy of the $T_m$ estimates. For each station, the regression of rainy-day model has a higher precision than that of the rainless-day model. On average, the RMS of the regression of rainy-day model is 28% smaller than that of the regression of rainless-day model, suggesting that the $T_m$ of rainy days estimated by the rainy-day model is expected to be more accurate than that of rainless days estimated by the rainless-day model.

Another contribution of this study is the exploration of the weather backgrounds for radiosonde data satisfying $T_m - T_s > 10$ K and $T_s - T_m > 30$ K that most deviate from the regression of the $T_s \sim T_m$ linear model. Radiosonde data satisfying $T_m - T_s > 10$ K are related to the weather of low surface temperature (<270 K) and both striking temperature and humidity inversions occurring below 800 hPa, while radiosonde data satisfying $T_s - T_m > 30$ K are related to the weather of high surface temperature (>280 K) and a striking humidity inversion normally occurring above 600 hPa without temperature inversion at any observation height.

Since the calculation of $T_m$ is based on the vertical distribution of temperature and water vapor pressure, it is not difficult to understand that local weather and climate determine the value of $T_m$, the relationship between $T_m$ and $T_s$, and the regression precision of $T_s \sim T_m$ linear models. On the other hand, many evidences provided in this study suggest that some $T_m$-related information reflects the local weather and climate, such as the situation of $T_m - T_s > 10$ K reflecting the (temperature and humidity) inversions and larger RMS of the regression of $T_s \sim T_m$ linear model indicating the poor uniformity of atmospheric profiles. Thus, it is promising to use the $T_m$ as an indicator for weather and climate studies, which deserves further investigation.

**Author Contributions:** Conceptualization, M.W. and J.C.; Methodology, M.W. and J.H.; software, M.W. and Y.Z.; Validation M.F., M.Y. and C.S.; formal analysis, M.W.; writing—original draft prepa-

ration, M.W.; writing—review and editing, J.C. and T.X. All authors have read and agreed to the published version of the manuscript.

**Funding:** This research was funded by the Opening Project of Shanghai Key Laboratory of Space Navigation and Positioning Techniques (NO. 202103), the Open Fund of Key Laboratory of Marine Environmental Survey Technology and Application, Ministry of Natural Resources (NO. MESTA-2020-B011), and the Startup Foundation for Introducing Talent of NUIST (NO. 2020R053; 2022R118).

**Data Availability Statement:** The radiosonde data used in this study are available from http://weather.uwyo.edu/upperair/sounding.html (accessed on 1 May 2022). The daily precipitation data for stations in mainland China are available from http://data.cma.cn/ (accessed on 16 May 2022), while for station KINGSPARK, Hong Kong, the data are from http://www.hko.gov.hk/tc/cis/awsDailyExtract.htm?stn=KP (accessed on 20 May 2022).

**Acknowledgments:** The authors would like to thank three anonymous reviewers for their constructive comments and useful suggestions to improve the manuscript.

**Conflicts of Interest:** The authors declare no conflict of interest.

## Appendix A

**Table A1.** Radiosonde station information and site-specific $T_s \sim T_m$ linear models. Columns 1 (C1)–4 (C4) present the information of the 88 radiosonde stations. Column 1 shows the series number of the stations. Column 2 shows the station ID and station name. Column 3 shows the latitude and longitude of the stations. Column 4 shows the altitude of the stations. The site-specific $T_s \sim T_m$ linear models of each station include a weather-independent model (generated from all radiosonde data), a rainless-day model (generated from the data of rainless days), and a rainy-day model (generated from the data of rainy days). Columns 5–10 present the linear regression coefficients of these models.

| Radiosonde Station | | | | Weather-Indenpedent Model | | Rainless-Day Model | | Rainy-Day Model | |
|---|---|---|---|---|---|---|---|---|---|
| NO. | ID/STN | Lat/Lon | H (m) | a | b | a | b | a | b |
| C1 | C2 | C3 | C4 | C5 | C6 | C7 | C8 | C9 | C10 |
| 1 | 45004 Kings Park | 22.31 114.16 | 66 | 0.57 | 118.16 | 0.62 | 102.43 | 0.49 | 139.34 |
| 2 | 50527 Hailar | 49.21 119.75 | 611 | 0.72 | 69.55 | 0.71 | 72.53 | 0.76 | 56.88 |
| 3 | 50557 Nenjiang | 49.16 125.23 | 243 | 0.77 | 55.79 | 0.75 | 59.88 | 0.81 | 42.53 |
| 4 | 50774 Yichun | 47.71 128.9 | 232 | 0.83 | 38.75 | 0.81 | 44.72 | 0.88 | 25.43 |
| 5 | 50953 Harbin | 45.75 126.76 | 143 | 0.85 | 33.30 | 0.85 | 34.35 | 0.87 | 27.88 |
| 6 | 51076 Altay | 47.73 88.08 | 737 | 0.65 | 90.53 | 0.63 | 96.10 | 0.70 | 75.13 |
| 7 | 51431 Yining | 43.95 81.33 | 664 | 0.65 | 89.68 | 0.61 | 102.63 | 0.74 | 63.36 |
| 8 | 51463 Urumqi | 43.78 87.62 | 919 | 0.60 | 103.13 | 0.57 | 111.42 | 0.64 | 90.21 |
| 9 | 51644 Kuqa | 41.71 82.95 | 1100 | 0.62 | 97.46 | 0.61 | 100.22 | 0.70 | 74.88 |
| 10 | 51709 Kashi | 39.46 75.98 | 1291 | 0.60 | 102.14 | 0.58 | 109.11 | 0.69 | 77.31 |

**Table A1.** *Cont.*

| | Radiosonde Station | | | Weather-Indenpedent Model | | Rainless-Day Model | | Rainy-Day Model | |
|---|---|---|---|---|---|---|---|---|---|
| NO. | ID/STN | Lat/Lon | H (m) | a | b | a | b | a | b |
| C1 | C2 | C3 | C4 | C5 | C6 | C7 | C8 | C9 | C10 |
| 11 | 51777 Ruoqiang | 39.03 88.16 | 889 | 0.58 | 109.52 | 0.57 | 111.79 | 0.67 | 84.11 |
| 12 | 51828 Hotan | 37.13 79.93 | 1375 | 0.61 | 97.69 | 0.60 | 101.21 | 0.69 | 75.81 |
| 13 | 51839 Minfeng | 37.06 82.71 | 1409 | 0.58 | 108.82 | 0.56 | 112.92 | 0.69 | 77.39 |
| 14 | 52203 Hami | 42.81 93.51 | 739 | 0.62 | 98.06 | 0.61 | 99.81 | 0.68 | 79.26 |
| 15 | 52267 Ejin Qi | 41.95 101.06 | 941 | 0.64 | 92.16 | 0.63 | 92.55 | 0.65 | 88.61 |
| 16 | 52323 Maz. Shan | 41.80 97.03 | 1770 | 0.64 | 89.84 | 0.63 | 93.16 | 0.72 | 68.73 |
| 17 | 52418 Dunhuang | 40.15 94.68 | 1140 | 0.59 | 105.47 | 0.58 | 108.71 | 0.71 | 69.68 |
| 18 | 52533 Jiuquan | 39.76 98.48 | 1478 | 0.62 | 96.06 | 0.60 | 101.51 | 0.72 | 67.19 |
| 19 | 52681 Minqin | 38.63 103.08 | 1367 | 0.65 | 89.38 | 0.63 | 93.73 | 0.74 | 61.29 |
| 20 | 52818 Golmud | 36.41 94.90 | 2809 | 0.60 | 98.88 | 0.58 | 104.50 | 0.69 | 74.83 |
| 21 | 52836 Dulan | 36.30 98.10 | 3192 | 0.75 | 57.22 | 0.73 | 63.54 | 0.80 | 46.86 |
| 22 | 52866 Xining | 36.71 101.75 | 2296 | 0.66 | 86.40 | 0.62 | 96.60 | 0.78 | 53.66 |
| 23 | 52983 Yu Zhong | 35.87 104.15 | 1875 | 0.67 | 84.57 | 0.64 | 92.16 | 0.77 | 55.08 |
| 24 | 53068 Erenhot | 43.65 112.00 | 966 | 0.68 | 78.05 | 0.67 | 81.56 | 0.75 | 60.08 |
| 25 | 53463 Hohhot | 40.81 111.68 | 1065 | 0.77 | 54.02 | 0.76 | 57.16 | 0.82 | 38.47 |
| 26 | 53513 Linhe | 40.76 107.40 | 1041 | 0.76 | 59.71 | 0.76 | 59.63 | 0.81 | 45.27 |
| 27 | 53614 Yinchuan | 38.48 106.21 | 1112 | 0.74 | 63.52 | 0.74 | 65.35 | 0.79 | 49.17 |
| 28 | 53772 Taiyuan | 37.78 112.55 | 779 | 0.77 | 54.03 | 0.77 | 55.92 | 0.82 | 41.52 |
| 29 | 53845 Yan An | 36.60 109.50 | 959 | 0.72 | 68.70 | 0.71 | 74.04 | 0.81 | 45.08 |
| 30 | 53915 Pingliang | 35.55 106.66 | 1348 | 0.77 | 56.72 | 0.75 | 61.35 | 0.83 | 38.31 |
| 31 | 54102 Xilin Hot | 43.95 116.06 | 991 | 0.74 | 64.15 | 0.71 | 70.49 | 0.80 | 45.91 |

**Table A1.** *Cont.*

| | Radiosonde Station | | | Weather-Indenpedent Model | | Rainless-Day Model | | Rainy-Day Model | |
|---|---|---|---|---|---|---|---|---|---|
| NO. | ID/STN | Lat/Lon | H (m) | a | b | a | b | a | b |
| C1 | C2 | C3 | C4 | C5 | C6 | C7 | C8 | C9 | C10 |
| 32 | 54135 Tongliao | 43.60 122.26 | 180 | 0.88 | 24.00 | 0.88 | 23.01 | 0.88 | 22.81 |
| 33 | 54161 Changchun | 43.90 125.21 | 238 | 0.87 | 27.66 | 0.87 | 27.91 | 0.88 | 24.66 |
| 34 | 54218 Chifeng | 42.26 118.96 | 572 | 0.85 | 32.86 | 0.84 | 34.16 | 0.86 | 28.66 |
| 35 | 54292 Yanji | 42.88 129.46 | 178 | 0.92 | 13.44 | 0.93 | 12.03 | 0.93 | 11.70 |
| 36 | 54342 Shenyang | 41.76 123.43 | 43 | 0.82 | 41.59 | 0.82 | 41.50 | 0.85 | 31.26 |
| 37 | 54374 Linjiang | 41.71 126.91 | 333 | 0.82 | 41.93 | 0.82 | 44.82 | 0.87 | 27.26 |
| 38 | 54511 Beijing | 39.93 116.28 | 55 | 0.87 | 25.21 | 0.87 | 24.89 | 0.86 | 28.63 |
| 39 | 54662 Dalian | 38.90 121.63 | 97 | 0.96 | 2.57 | 0.97 | -0.20 | 0.93 | 10.08 |
| 40 | 54727 Zhangqiu | 36.70 117.55 | 123 | 0.83 | 37.24 | 0.84 | 36.95 | 0.82 | 40.26 |
| 41 | 54857 Qingdao | 36.06 120.33 | 77 | 0.96 | 3.35 | 0.98 | -3.63 | 0.91 | 17.59 |
| 42 | 55299 Nagqu | 31.48 92.06 | 4508 | 0.67 | 77.83 | 0.55 | 109.67 | 0.75 | 58.35 |
| 43 | 55591 Lhasa | 29.66 91.13 | 3650 | 0.63 | 93.57 | 0.60 | 101.16 | 0.67 | 82.90 |
| 44 | 56029 Yushu | 33.01 97.01 | 3682 | 0.73 | 64.51 | 0.66 | 81.31 | 0.76 | 56.47 |
| 45 | 56080 Hezuo | 35.00 102.90 | 2910 | 0.74 | 63.76 | 0.69 | 76.10 | 0.82 | 42.92 |
| 46 | 56137 Qamdo | 31.15 97.16 | 3307 | 0.70 | 74.10 | 0.65 | 87.75 | 0.71 | 70.39 |
| 47 | 56146 Garze | 31.61 100.00 | 522 | 0.72 | 70.94 | 0.66 | 86.30 | 0.76 | 58.98 |
| 48 | 56187 Wenjiang | 30.70 103.83 | 541 | 0.71 | 72.29 | 0.69 | 80.93 | 0.79 | 48.56 |
| 49 | 56571 Xichang | 27.90 102.26 | 1599 | 0.58 | 110.66 | 0.55 | 121.73 | 0.70 | 75.35 |
| 50 | 56691 Weining | 26.86 104.28 | 2236 | 0.62 | 102.15 | 0.60 | 107.06 | 0.60 | 105.10 |
| 51 | 56739 Tengchong | 25.11 98.48 | 1649 | 0.52 | 130.75 | 0.53 | 127.02 | 0.61 | 102.11 |
| 52 | 56778 Kunming | 25.01 102.68 | 1892 | 0.45 | 148.44 | 0.41 | 161.20 | 0.62 | 99.25 |

**Table A1.** *Cont.*

| | Radiosonde Station | | | Weather-Indenpedent Model | | Rainless-Day Model | | Rainy-Day Model | |
|---|---|---|---|---|---|---|---|---|---|
| NO. | ID/STN | Lat/Lon | H (m) | a | b | a | b | a | b |
| C1 | C2 | C3 | C4 | C5 | C6 | C7 | C8 | C9 | C10 |
| 53 | 56964 Simao | 22.76 100.98 | 1303 | 0.35 | 181.68 | 0.35 | 182.07 | 0.54 | 124.71 |
| 54 | 56985 Mengzi | 23.38 103.38 | 1302 | 0.49 | 138.31 | 0.49 | 141.13 | 0.58 | 113.11 |
| 55 | 57083 Zhengzhou | 34.71 113.65 | 111 | 0.81 | 46.17 | 0.81 | 45.67 | 0.81 | 44.03 |
| 56 | 57127 Hanzhong | 33.06 107.03 | 509 | 0.78 | 54.11 | 0.75 | 61.72 | 0.86 | 27.99 |
| 57 | 57131 Jinghe | 34.43 108.97 | 411 | 0.75 | 60.92 | 0.74 | 64.27 | 0.83 | 39.08 |
| 58 | 57178 Nanyang | 33.03 112.58 | 131 | 0.80 | 48.24 | 0.80 | 48.23 | 0.82 | 42.57 |
| 59 | 57447 Enshi | 30.28 109.46 | 458 | 0.77 | 56.77 | 0.74 | 66.97 | 0.81 | 45.32 |
| 60 | 57461 Yichang | 30.70 111.30 | 134 | 0.80 | 48.09 | 0.81 | 47.18 | 0.80 | 48.65 |
| 61 | 57494 Wuhan | 30.61 114.13 | 23 | 0.75 | 64.14 | 0.75 | 63.45 | 0.74 | 66.97 |
| 62 | 57516 Chongqing | 29.51 106.48 | 260 | 0.81 | 47.57 | 0.77 | 58.23 | 0.83 | 38.87 |
| 63 | 57679 Changsha | 28.20 113.08 | 46 | 0.70 | 78.60 | 0.71 | 76.15 | 0.67 | 85.54 |
| 64 | 57749 Huaihua | 27.56 110.00 | 261 | 0.68 | 83.41 | 0.68 | 83.76 | 0.67 | 86.91 |
| 65 | 57816 Guiyang | 26.48 106.65 | 1222 | 0.62 | 101.20 | 0.64 | 94.89 | 0.59 | 108.71 |
| 66 | 57957 Guilin | 25.33 110.30 | 166 | 0.63 | 99.24 | 0.67 | 87.73 | 0.59 | 110.69 |
| 67 | 57972 Chenzhou | 25.80 113.03 | 185 | 0.62 | 101.09 | 0.64 | 96.77 | 0.60 | 107.27 |
| 68 | 57993 Ganzhou | 25.85 114.95 | 125 | 0.63 | 98.02 | 0.64 | 96.53 | 0.62 | 102.24 |
| 69 | 58027 Xuzhou | 34.28 117.15 | 42 | 0.84 | 37.63 | 0.85 | 35.50 | 0.83 | 38.18 |
| 70 | 58150 Sheyang | 33.76 120.25 | 7 | 0.86 | 32.34 | 0.87 | 27.96 | 0.86 | 29.31 |
| 71 | 58203 Fuyang | 32.86 115.73 | 33 | 0.84 | 37.86 | 0.85 | 35.13 | 0.81 | 44.24 |
| 72 | 58238 Nanjing | 32.00 118.80 | 7 | 0.81 | 44.31 | 0.82 | 42.58 | 0.81 | 46.42 |
| 73 | 58362 Shanghai | 31.40 121.46 | 4 | 0.82 | 42.85 | 0.82 | 41.29 | 0.81 | 43.50 |

**Table A1.** *Cont.*

| | Radiosonde Station | | | Weather-Indenpedent Model | | Rainless-Day Model | | Rainy-Day Model | |
|---|---|---|---|---|---|---|---|---|---|
| NO. | ID/STN | Lat/Lon | H (m) | a | b | a | b | a | b |
| C1 | C2 | C3 | C4 | C5 | C6 | C7 | C8 | C9 | C10 |
| 74 | 58424 Anqing | 30.53 117.05 | 20 | 0.79 | 51.17 | 0.81 | 45.75 | 0.76 | 61.78 |
| 75 | 58457 Hangzhou | 30.23 120.16 | 43 | 0.79 | 50.87 | 0.80 | 49.28 | 0.78 | 53.00 |
| 76 | 58606 Nanchang | 28.60 115.91 | 50 | 0.74 | 67.77 | 0.76 | 61.88 | 0.70 | 78.62 |
| 77 | 58633 Qu Xian | 28.96 118.86 | 71 | 0.73 | 70.12 | 0.72 | 73.11 | 0.74 | 67.51 |
| 78 | 58665 Hongjia | 28.61 121.41 | 2 | 0.78 | 54.31 | 0.80 | 50.01 | 0.77 | 58.05 |
| 79 | 58725 Shaowu | 27.33 117.46 | 219 | 0.69 | 81.32 | 0.69 | 82.75 | 0.71 | 76.99 |
| 80 | 58847 Fuzhou | 26.08 119.28 | 85 | 0.73 | 69.83 | 0.75 | 64.35 | 0.70 | 78.76 |
| 81 | 59134 Xiamen | 24.48 118.08 | 139 | 0.68 | 84.50 | 0.73 | 71.76 | 0.60 | 108.31 |
| 82 | 59211 Baise | 23.90 106.60 | 175 | 0.64 | 95.58 | 0.64 | 97.28 | 0.65 | 93.27 |
| 83 | 59265 Wuzhou | 23.48 111.30 | 120 | 0.58 | 114.12 | 0.62 | 104.13 | 0.54 | 125.68 |
| 84 | 59280 Qing Yuan | 23.66 113.05 | 19 | 0.59 | 111.25 | 0.64 | 97.52 | 0.54 | 126.00 |
| 85 | 59316 Shantou | 23.35 116.66 | 3 | 0.63 | 100.81 | 0.67 | 87.97 | 0.56 | 120.00 |
| 86 | 59431 Nanning | 22.63 108.21 | 126 | 0.54 | 125.08 | 0.58 | 114.61 | 0.50 | 138.79 |
| 87 | 59758 Haikou | 20.03 110.35 | 24 | 0.56 | 121.84 | 0.61 | 108.11 | 0.50 | 137.63 |
| 88 | 59981 Xisha Dao | 16.83 112.33 | 5 | 0.47 | 149.53 | 0.41 | 167.98 | 0.47 | 147.96 |

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
