# Peer review of "Region-Specific and Weather-Dependent Characteristics of the Relation between GNSS-Weighted Mean Temperature and Surface Temperature over China"

_remotesensing, doi:10.3390/rs15061538_

Round 1

Author Response

Dear reviewer 1,

Please see the attached response letter.

Best regards,

Minghua Wang

Reviewer 2 Report

The abstract need to be modified, it does not correct put an acronym.

Line 58-59: specify the trouble in practical applications.

Please check the whole equation.

Why are you using 14 years of data? Explain more, please

Map, please add labels such as country and seas, remove the small map is not necessary. Add a legend to the map. Please add the reference of GMT and the relief model used in the map caption. All maps need to be modified.

What about “we carried out the quality checking for each radiosonde profile.”? Please explain more. What did you find?

“This suggests that, in general, using weather-dependent models for rainy days and no rain days separately will effectively improve the accuracy in the west and north of China, but not at the east part.” What happens with some red points in the middle of blue points? Could it be another reason?

Author Response

Dear reviewer 2,
Please see the attached response letter.
Best regards,

Minghua Wang

Reviewer 3 Report

Authors presents an interesting study in the manuscript entitled «Region-specific and weather-dependent characteristics of the relation between GNSS weighted mean temperature and surface temperature over China».  After detailed analyzing the manuscript we may note a few points to be corrected.

I)                   Figure 4. It is not quite clear how you chose the sequence number of the stations shown in the figure 4.  Please, clarify this question. Where are the stations 10,20, 30, 40 located.

II)                It is possible to improve manuscript if you connect  two cases Tm-Ts>10 K and Ts-Tm>30 K not only with the weather of low surface temperature and the weather of high surface temperature but also with concrete atmospheric conditions (cold or warm weather fronts, influence of local atmospheric circulations,  evolution of atmospheric pressure large scale structures and so on). It will be useful as you consider large region.

III)             In some places of the manuscript there are incorrect relation Ts ? Tm (for example, lines 441, 450).

IV)             Please clarify, do you connect the largest differences (figure 6) with influence of the lower layers of the atmosphere?  Does the vertical resolution somehow affect the explanation of the observed differences?

V)                It is necessary to expand introduction and discuss the fact that estimations of precipitable water vapor from GNSS measurement are close to the ERA-5 and radiosonde data. You may use the following papers:

- Shikhovtsev, A.Y., Khaikin, V.B., Mironov, A.P. et al. Statistical Analysis of the Water Vapor Content in North Caucasus and Crimea. Atmos Ocean Opt 35, 168–175 (2022). https://doi.org/10.1134/S1024856022020105.

-S. Z. Ziv, Y. Yair, P. Alpert, L. Uzan, and Y. Reuveni, “The diurnal variability of precipitable water vapor derived from GPS tropospheric path delays over the Eastern Mediterranean,” Atmos. Res. 249, 105307 (2021).

-Y. Zhang, C. Cai, B. Chen, and W. Dai, “Consistency evaluation of precipitable water vapor derived from ERA5, ERA-Interim, GNSS, and radiosondes over China,” Radio Sci. 54, 561–571 (2019).

In whole the manuscript is very interesting. Besides the recommendations discussed above we also advice to  authors to emphasize the novelty of your research. After corrections we may recommend the manuscript for publication.

Author Response

(The authors gave the same response as above.)

Round 2

Reviewer 3 Report

We agree with your approach and corrections.